# NK Cell Degranulation Triggered by Rituximab Identifies Potential Markers of Subpopulations with Enhanced Cytotoxicity toward Malignant B Cells

**DOI:** 10.3390/ijms25168980

**Published:** 2024-08-18

**Authors:** Marta Wlodarczyk, Anna Torun, Abdessamad Zerrouqi, Beata Pyrzynska

**Affiliations:** 1Chair and Department of Biochemistry, Faculty of Medicine, Medical University of Warsaw, 02-091 Warsaw, Poland; marta.wlodarczyk@wum.edu.pl (M.W.); a.torun@wp.pl (A.T.); abdessamad.zerrouqi@wum.edu.pl (A.Z.); 2Department of Biochemistry and Pharmacogenomics, Faculty of Pharmacy, Medical University of Warsaw, 02-097 Warsaw, Poland; 3Centre for Preclinical Research, Medical University of Warsaw, 02-097 Warsaw, Poland; 4Institute of Mother and Child, 01-211 Warsaw, Poland

**Keywords:** NK cells, CD16, Burkitt lymphoma, mAbs, rituximab, c-KIT, OX40L

## Abstract

A promising strategy in cancer immunotherapy is to restore or enhance the cytotoxicity of NK cells, among others, by activating the mechanism of antibody-dependent cellular cytotoxicity (ADCC). Monoclonal antibodies targeting tumor antigens, such as rituximab (targeting CD20), induce NK cell-mediated ADCC and have been used to treat B cell malignancies, such as non-Hodgkin lymphoma, but not always successfully. The aim of this study was to analyze the gene expression profile of the NK cells involved in the cytolytic response stimulated by rituximab. NK cells were co-cultured with rituximab-opsonized Raji cells. Sorting into responder and non-responder groups was based on the presence of CD107a, which is a degranulation marker. RNA-seq results showed that the *KIT* and *TNFSF4* genes were strongly down-regulated in the degranulating population of NK cells (responders); this was further confirmed by qRT-PCR. Both genes encode surface proteins with cellular signaling abilities, namely c-KIT and the OX40 ligand. Consistent with our findings, c-KIT was previously reported to correlate inversely with cytokine production by activated NK cells. The significance of these findings for cancer immunotherapy seems essential, as the pharmacological inhibition of c-KIT and OX40L, or gene ablation, could be further tested for the enhancement of the anti-tumor activity of NK cells in response to rituximab.

## 1. Introduction

In recent years, significant progress has been made towards understanding the critical role of natural killer (NK) cells in the immune response to tumors. These cytotoxic lymphocytes, with their inherent ability to identify and eliminate stressed cells, including tumor cells and specific infected cells, offer a promising avenue in cancer treatment. Some studies have highlighted a correlation between low NK cell activity and an increased risk of cancer [1]. As a result, there is a growing focus on enhancing NK cell function through various therapeutic approaches, including mobilizing endogenous NK cells with treatments, introducing ex vivo-expanded NK cell populations, or engineering NK cells with chimeric antigen receptors as a form of cellular therapy.

One promising avenue for boosting NK cell activity in treating hematologic and solid tumors is the use of therapeutic tumor-targeting monoclonal antibodies (mAbs). These mAbs, with their remarkable efficacy, have the potential to significantly enhance the immune response against cancer [2,3]. Notably, rituximab (RTX), a chimeric monoclonal antibody targeting the CD20 antigen on B cells, has dramatically improved the prognosis and the outcomes in various B cell malignancies such as non-Hodgkin lymphoma (B-NHL) and chronic lymphocytic leukemia (CLL) by depleting CD20-positive B cells [3,4]. In fact, the mechanism of response to rituximab is complex. The binding of RTX to CD20 induces cell death through various mechanisms involving the patient’s immune system [5], including antibody-dependent cell-mediated cytotoxicity (ADCC), complement-dependent cytotoxicity (CDC), and antibody-dependent phagocytosis (ADP) (Figure 1).

ADCC, however, is thought to be a significant contributor to the in vivo antitumor activity of RTX. The binding of the variable region of the therapeutic mAb to CD20 facilitates the binding of its Fc region to FcγRIII on NK cells, further leading to the formation of the immune synapse and triggering a response in cytotoxic NK cells to release granules containing perforin and granzyme B, which subsequently induce the cell death of the target cancer cell. CDC is mediated by the binding of the C1 complex to the Fc fragment of RTX bound to opsonized cells, which triggers the complement cascade, resulting in the insertion of the membrane attack complex (MAC) into the target cell membrane, leading ultimately to the target cell lysis [6]. ADP is the least studied rituximab effector mechanism with no existing in vivo evidence of RTX-mediated ADP in humans. Macrophages recognize CD20-bound RTX through various Fcγ receptors, which leads to the antibody-dependent phagocytosis (ADP) of the target cell. The contribution of this mechanism during anti-CD20 therapy remains limited [7]. An additional mechanism, which is yet to be disputed, has been proposed where the binding of RTX to CD20 causes a cross-linking of multiple CD20 molecules and triggers a non-classical apoptotic program in RTX-opsonized cells [8]. While rituximab has been a breakthrough in cancer treatment, it is important to recognize that some patients do not respond adequately or develop resistance to it over time. In the case of DLBCL, 30–50% of patients are not cured by R-CHOP (RTX-based therapy), with about 20% being initially refractory and another 30% relapsing after a complete response [2,3,9]. This underscores the urgent need to understand and overcome these challenges. The effectiveness of RTX hinges on the interaction between the antibody and the NK cells, mainly through the low-affinity Fc-γ-receptor IIIa (FcγRIIIa; CD16) [10]. CD16 engaged with antibody-coated target cells can induce a full display of effector functions and, subsequently, degranulation by NK cells [11] and phagocytosis by macrophages at the proximity of tumor cells [7]. CD16 signaling depends on the participation of many proteins within the CD16 complex cascades (Figure 2) through the phosphorylation of immunoreceptor tyrosine-based activation motifs (ITAMs) [12,13]. The signaling through ITAMs induces the activation of Syk family kinases [14], which in turn activate phospholipase C-gamma (PLCγ), which generates diacylglycerol (DAG) and inositol-1,4,5-trisphosphate (IP3) [13]. DAG activates protein kinase C (PKC), which in turn activates the transcription factor NF-κB, leading to the production of TNF-alpha (TNFα) [15]. The generated IP3 leads to the activation of the calcium/calmodulin-dependent phosphatase, calcineurin, and an increase in intracellular calcium levels [16]. The Syk-PLCγ1-Ca^2+^ signaling pathway triggered by CD16 binding leads to the rapid mobilization and exocytosis of cytotoxic granules containing lytic enzymes, such as perforin and granzymes, which are released into the immunological synapse formed between the NK cell and the target cell [13,16,17]. In addition to NK cell degranulation, calcineurin can dephosphorylate the transcription factor NFAT, allowing it to translocate to the nucleus and induce the expression of IFN-γ [17]. Activating transcription factors such as NF-κB and AP-1 further promote the synthesis and secretion of proinflammatory cytokines. The released cytokines are crucial in orchestrating the immune response by enhancing cytotoxicity and modulating adaptive immune responses.

The factors influencing NK cell reactivity to RTX are multifaceted, ranging from the features of target cells, such as the levels of expression or mutations of the CD20 antigen, to the immune suppressive factors in the microenvironment. With respect to target cells, CD20 expression levels are low in chronic lymphocytic leukemia (CLL), while the highest levels were observed in diffuse large B cell lymphoma (DLBCL) patients [18]. The efficacy of RTX in recognizing CD20 is influenced by the conformation of CD20 on the surface of malignant cells, which is altered by point mutations and deletions. The presence of point mutations in the *MS4A1* gene encoding CD20, such as (S97F; TCC→TTC) and (V247I; GTT→ATT), has been reported as being associated with decreased CD20 expression after RTX therapy in B cell lymphoma [19]. Mutations in the cytoplasmic C terminal of CD20 involved in the recognition by rituximab are rare in DLBCL and are not a significant cause of R-CHOP failure [20]. However, C-terminal deletion mutations in CD20 have been reported as being associated with the loss of CD20 expression in RTX-resistant non-Hodgkin’s lymphoma [21]. These CD20 mutations involving the RTX epitope are less present in DLBCL, but are still able to cause irreversible resistance to RTX [20].

The anti-tumor NK cell response can also be attributed to the factors associated with NK cells, such as a decreased expression of Fcγ receptors on NK cells. CD16 down-modulation represents a well-known mechanism under conditions of chronic CD16 engagement. This down-regulation of CD16 leads to a reduced ability to execute ADCC by either receptor internalization or disintegrin and metalloproteinase (ADAM)17-dependent shedding in the scenario of the persistent contact of NK cells with mAb-opsonized cells [22,23]. Along this line, the ex vivo analysis of RTX-treated DLBCL patients revealed a marked and prolonged therapy-induced reduction in both “natural” and CD16-dependent NK cytotoxic activities, accompanied by the down-modulation of CD16 and NKG2D, which is the activating receptor [24]. Acting separately or together, these factors could lead to resistance to therapeutic mAbs, a substantial decrease in therapeutic efficacy, and a high relapse rate [25]. Further studies have shown that combining an anti-CD20 antibody with Monalizumab, an antibody that efficiently releases the inhibition conferred by the engagement of the inhibitory receptor NKG2A on NK cells, enhances NK cell-mediated ADCC, emphasizing the importance of the activation status of NK cells in determining the efficacy of tumor-targeting mAbs [26]. Understanding the mechanisms behind NK cell response to RTX to overcome resistance to death by NK cells is crucial for maximizing the therapeutic benefits of mAbs.

This study examined the gene expression profiles of NK cells engaged in cytolytic activities triggered by the anti-CD20 antibody RTX. After exposure to RTX-opsonized Raji cells, the degranulation of NK cells was used as a marker to distinguish responding versus non-responding NK cells. The differences in the gene expression levels allowed us to gain valuable insights into the potential factors involved in NK cell resistance and the response against tumor cells treatment with RTX.

## 2. Results

### 2.1. NK Cells Exposed to Rituximab-Coated Lymphoma Cells Degranulate Non-Uniformly

To characterize the CD16 receptor-mediated responses of donor-derived primary NK cells to therapy with monoclonal antibodies, we conducted degranulation-type experiments (Figure 3). To stimulate the CD16-mediated activation of NK cells, we incubated them with the target Burkitt lymphoma cell line Raji, pre-coated with the anti-CD20 therapeutic monoclonal antibody RTX. It is expected that under such conditions, the lytic granules of NK cells, containing granzymes, perforin, and glycoprotein CD107a/LAMP1 (Lysosomal Associated Membrane Protein 1), move to the vicinity of the target cell. The CD107a protein is known for its participation in perforin transport and cytotoxic granule movement to the cell surface [27]. Upon degranulation, the CD107a becomes exposed on the surface of the NK cell.

Therefore, the binding of the anti-CD107a antibody (degranulation marker) to NK cells allowed us to identify, using flow cytometry, almost equally abundant populations of NK cells, the degranulating (“responders”), and non-degranulating (“non-responders”) in NK cell populations isolated from the PBMCs of six healthy donors (Table 1).

Figure 4 presents the gating strategy for the flow cytometry analysis of degranulating NK cells.

Additionally, to confirm that the observed degranulation in the responder population of NK cells was indeed a result of their activation, we performed the detection of intracellular TNFα (one of the proinflammatory cytokines produced by NK cells early upon activation), which enhances NK cell function and proliferation [28,29]. Co-incubation with RTX-coated Raji cells resulted in about 29–38% of the total population of NK cells producing increased levels of TNFα (Figure 5A,B). Notably, the mean fluorescence intensity (MFI) corresponding to the intensity of the TNFα signal was increased upon incubation with RTX-coated Raji in the responder’s population almost four times compared to non-responders (Figure 5C). Cumulatively, these results demonstrated that in response to RTX-coated lymphoma cells, the NK cells are activated non-uniformly, in the matter of both cytokine production and degranulation.

### 2.2. Analysis of CD16-Dependent Transcriptional Changes in Primary NK Cells

To analyze the CD16 signaling-mediated changes in gene expression, we separately sorted the non-responder and the responder populations of NK cells derived from the healthy donors using the flow cytometry sorter and gating strategy presented in Figure 4. The quality of total RNA isolated from these sorted NK cells was initially checked by electrophoresis on agarose gel. This allowed for the estimation of its integrity by means of the presence of sharp bands of 28S and 18S rRNA and the lack of an RNA smear. Additionally, the QC report provided by Omega Bioservices company (Norcross, GA, USA) confirmed that all RNA samples were of high quality, with the RIN (RNA Integrity Number) values ranging between 9.8 and 10.0, and passed the QC test. An equal amount of RNA was further processed for RNA sequencing (RNA-seq) using the next-generation sequencing (NGS) method and the HiSeq X Ten platform, followed by the bioinformatic analysis serviced by Omega Bioservices company. The differential expression analysis (Figure 6A,B and Appendix A) revealed three mRNAs significantly down-regulated (*p*-value < 0.05; log2 of fold change < −0.6), namely *KIT*, *S1PR5*, and *TNFSF4*. Using the same criteria, only one mRNA, *SIPA1L2*, was significantly up-regulated (*p*-value < 0.05; log2 of fold change > 0.6).

### 2.3. KIT and TNFSF4 Genes Are Down-Regulated upon CD16 Stimulation

The down-regulation of gene expression appeared to be the main transcriptional change detected in degranulating responder NK cells compared to non-responders. To validate our results, we performed a quantitative analysis of *KIT* and *TNFSF4* mRNAs using qRT-PCR, as these mRNAs exhibited the most significant down-regulation, with the highest magnitude in the responder group (Figure 6A). The responder and the non-responder populations of NK cells were FACS-sorted again from the independent ADCC experiments performed with primary NK cells derived from three different donors than those that were previously used. The results of qRT-PCR (Figure 6C) confirmed the very consistent down-regulation of both mRNAs in the responder population of NK cells derived from all three donors, compared to the non-responder population.

## 3. Discussion

NK cells contribute to anti-tumor immunity by performing immunosurveillance and eliminating pre-cancerous cells. Importantly, NK cells are also essential executors of anti-cancer immunotherapies, such as therapies with mAbs or chimeric antigen receptors (CARs). CD16-mediated signaling in NK cells is critical to the outcome response to treatment with monoclonal antibodies. CD16 recognizes the Fc fragment of antibodies, leading to the activation of NK cells, the release of various cytokines or chemokines, and the NK cells’ degranulation. In our experiments, we induced CD16 activation using the therapeutic anti-CD20 monoclonal antibody RTX, bound to the target antigen-expressing Raji cells, leading to the NK cells’ degranulation. We measured the production of TNFα cytokine and the degranulation of NK cells using flow cytometry, as the NK cells that degranulated exposed the CD107a antigen on their surface. To obtain insight into changes in gene expression upon the activation of CD16 signaling, we sorted the CD107a-positive (responders) and -negative (non-responders) NK cells and compared their gene expression profiling. The results of RNA-seq analysis pointed to the strong down-regulation of a few particular genes in the responder’s population compared to non-responders, with the highest magnitude of down-regulation exhibited by the *KIT* and *TNFSF4* genes. The quantification of gene expression with the RT-PCR method, using samples isolated from separate NK cell donors, confirmed that, indeed, both genes were strongly down-regulated in the responder population of NK cells, which had already degranulated. There are some literature reports focusing on the role of *KIT* and *TNFSF4* in the anti-tumor activity of NK cells.

The *KIT* gene encodes the transmembrane receptor tyrosine kinase c-KIT (CD117), which was originally associated with a role in NK cell development and maturation [30,31]. The c-KIT receptor is activated by the stem cell factor (SCF), providing signals for the expansion of NK-committed precursors and NK cell survival and final maturation [32]. However, very few studies have specifically addressed the role of c-KIT in NK cell function. Interestingly, an inverse correlation has been observed between the interferon-gamma (IFN-γ) production and the c-KIT level in NK cells [33]. The production of cytokines, such as IFN-γ and TNFα, can be considered surrogates of NK cell activation. Therefore, it is not surprising that activated, degranulating NK cells exhibit both cytokine production and the down-regulation of *KIT* expression. However, the molecular mechanism underlying such inverse correlation has not been explored. In mesenchymal stem cells, the kinase c-KIT regulates the phosphorylation and activity of mTOR [34]. In NK cells, the kinase mTOR plays an essential role in cytokine secretion and promotes IFN-γ synthesis [33]. Therefore, Bosken et al. suggested that c-KIT may also negatively influence mTOR signaling in NK cells, as an inverse correlation between the level of c-KIT and mTOR phosphorylation has been found in NK cells activated by the exposure to heat-killed bacteria *S. aureus* [33]. In line with these findings, our results demonstrated a striking inverse correlation between CD16-mediated NK cell activation (measured by degranulation and TNFα production—the characteristics of the responder group) and *KIT* gene expression. However, further studies are needed to explore whether the NK cell’s activation led to the *KIT* down-regulation or whether the NK cells with lower *KIT* expression were better predisposed to activation and degranulation, which qualified them as members of the responder group.

The *TNFSF4* gene encodes a tumor necrosis factor superfamily member, the OX40 ligand (OX40L), which is known to increase NK cell expansion and promote NK cell function by activating the OX40 cascade [35,36,37]. The OX40 activation cascade is well characterized in T cells, where it leads to the recruitment of the adaptor proteins TRAFs, followed by the stimulation of the transcription factors NF-κB and NFAT. In addition, OX40L can itself transduce the intracellular signals, similarly to other TNF family members [38]. OX40L signaling leads to the enhancement of NK cell cytotoxicity directed toward AML cells [39]. The expression of the *TNFSF4* gene also correlated with the response of recurrent glioblastoma patients to autologous NK cell therapeutics, serving as a potential biomarker of responsiveness [40]. However, its expression in the glioblastoma tissue of patients responding to the therapy was associated with the presence of M2-type macrophages rather than to NK cells. Additionally, a higher expression of *TNFSF4* in T cells, not NK cells themselves, contributed to better NK cell migration toward cancer cells in assays in vitro [40]. Consistently, the low expression of *TNFSF4* correlated with a worse prognosis and outcome of anti-PD1 therapy in patients with melanoma [41]. Therefore, at first glance, it seems paradoxical that the gene encoding the OX40L protein, essential for the expansion and cytotoxicity of NK cells and positively correlating with immunotherapy outcomes, is down-regulated upon NK cell activation via CD16 signaling. However, contradictory findings show that *TNFSF4* expression is associated with a poor prognosis in patients with diffuse gliomas [42]. Moreover, it has been found that the production of IFN-γ by NK cells is increased by the stimulation of OX40 by agonistic monoclonal antibodies, but not by OX40L [43].

Although the potential role of *KIT* and *TNFSF4* in regulating NK cell activity requires further studies, it seems striking that, specifically, these two genes are down-regulated in degranulating NK cells. Therefore, the low levels of c-KIT and OX40L could potentially be biomarkers of active NK cells. Moreover, the pharmacological inhibition of c-KIT [44,45], the antibody-based inhibition of OX40L [46], or genetic ablation could be a potential future strategy to improve the anti-tumor activity of NK cells in response to therapeutic treatment with monoclonal antibodies, such as RTX.

## 4. Materials and Methods

### 4.1. NK Cell Isolation

The buffy coats of healthy anonymous donors were obtained from the Blood Donation Center (Warsaw, Poland), followed by the isolation of peripheral blood mononuclear cells (PBMCs) using gradient centrifugation on the density gradient medium, Lymphoprep (STEMCELL Technologies, Vancouver, BC, Canada, cat. #07801), as per the manufacturer’s recommendations. Subsequently, the NK cells were isolated using the EasySep Human NK Cell Enrichment Kit (STEMCELL Technologies, cat. #19055). The NK cells were then suspended in RPMI 1640 medium (Corning, Corning, NY, USA, cat. #10-040-CVR) supplemented with 10% calf serum (Fisher Scientific, Waltham, MA, USA, cat. #SH3007203), penicillin/streptomycin solution (Sigma Aldrich, St. Louis, MO, USA, cat. #P4333), interleukin-2 (IL-2; 100 U/mL; Peprotech, Waltham, MA, USA, cat. #200-02), and interleukin-15 (IL-15; 10 ng/mL; Peprotech, cat. #200-15), and were cultured overnight in numerous wells of U-bottom 96-well plates (1.2 × 10^5^ cells in 100 μL medium/well).

### 4.2. Burkitt Lymphoma Cell Line Culture

The Raji cell line (RRID: CVCL_0511) was cultured in RPMI 1640 medium supplemented with 10% calf serum and penicillin/streptomycin solution (detailed in the previous paragraph). The cells regularly tested negative for mycoplasma contamination.

### 4.3. NK Cell Degranulation Analysis and Sorting

The target Raji cells were seeded in 96-well plates (1.2 × 10^5^ cells/well) and coated with RTX (10 μg/mL) via incubation for 30 min at 37 °C. The unbound RTX was removed by Raji cell centrifugation, followed by cell resuspension in 100 μL of complete RPMI 1640 medium supplemented with anti-CD107a-FITC antibody (BD Pharmingen, Franklin Lakes, NJ, USA, cat. #555800; 4 μL/well) and GolgiStop reagent (BD Bioscience, Franklin Lakes, NJ, USA, cat. #51-2092KZ; 0.8 μL/well). The effector NK cells were added (1.2 × 10^5^ cells/well) and co-incubated with Raji for 2 h at 37 °C. After centrifugation, the cells were stained with the mix of viability stain FVS510 (BD Bioscience, cat. #564406; diluted 200×) and anti-CD56-PE/Vio770 (Miltenyi Biotec, Gladbach, Germany, cat. #130-096-831; diluted 100×) for 20 min at 4 °C, followed by flow cytometry analysis using BD FACSAria III Cell Sorter (BD Bioscience). After the discrimination of singlets and viable cells (FVS510-negative events), the population of NK cells (CD56-positive events) was analyzed for degranulation marker CD107a. The populations of degranulating cells, “responders” (CD107a-positive events) and “non-responders” (CD107a-negative events), were sorted and collected in separate tubes. The precise localization of CD107a-negative events on the dot plots was assured by analyzing samples lacking RTX or Raji cells.

### 4.4. Intracellular TNFα Analysis

The experiments were initially performed as in the case of degranulation analysis. After 2 h of NK cell co-incubation with RTX-coated Raji, in the presence of anti-CD107a-FITC antibody, the cells were stained for 20 min with FVS510 and anti-CD56-PE/Vio770, as described above. After washing in PBS, the cells were resuspended in the Fixation/Permeabilization solution (BD Cytofix/Cytoperm Fixation/Permeabilization Kit (BD Biosciences, cat. #554714) and incubated for 20 min at 4 °C. After washing in BD Perm/Wash buffer, the samples were stained with anti-TNFα-eFluor 450 (eBioscience, San Diego, CA, USA, cat. #48-7349-41) for 30 min at 4 °C. Washed cells were analyzed using the BD FACSVerse Cell Analyzer (BD Bioscience).

### 4.5. RNA-Seq and Bioinformatics Analysis

RNA was isolated from the sorted NK cells (separated populations of responders and non-responders) using the RNeasy Plus Mini Kit (QIAGEN, Germantown, MD, USA, cat. #74134). The quantity and the integrity of RNA samples were initially estimated using the NanoDrop spectrophotometer and electrophoresis on ethidium bromide-stained 1% agarose gel, respectively. Further sample quantity, purity, and integrity assessment with the Agilent TapeStation system; RNA-seq library preparation with TruSeq Stranded mRNA Library kit (Illumina); library normalization; sequencing with HiSeq X Ten platform in a PE 2×150 bp format, with 10 million reads; and bioinformatics analysis (mapping, alignment, and differential expression analysis through Illumina’s Basespace software; https://basespace.illumina.com/, accessed on 8 February 2022) were commercially provided by Omega Bioservices company. The reference genome UCSC hg19, the BamStats summary method, Isis analysis software (version 2.6.25.19), BEDTools (version 2.17.0), Picard (version 1.128), and DESeq2 (version 1.6.3) were used. The annotation gene count was 26,364, while the assessed gene count was 10,222.

### 4.6. cDNA Synthesis and Quantitative Real-Time PCR

The total RNA isolated from NK cells was first treated with DNase I (Sigma-Aldrich, cat #AMPD1) to remove any remaining DNA, according to the manufacturer’s instructions. For cDNA synthesis, the reaction included 1 μg of DNase I-treated RNA, 1 mM dNTP mix, 2.5 μM random nonamers, and 2.5 μM oligo(dT)_23_ primers. After incubation at 70 °C for 10 min, the M-MLV reverse transcriptase was added and incubated at 37 °C for 50 min. To denature M-MLV transcriptase, the samples were incubated at 90 °C for 10 min. The random nonamers (cat. #R7647), oligo(dT)_23_ primers (cat. #O4387)_,_ and M-MLV reverse transcriptase (cat. #M1302) were all purchased from Sigma-Aldrich. The synthesized cDNA was diluted 10 times, and 2 μL was taken per PCR reaction. The LightCycler 480 II thermocycler (Roche, Basel, Switzerland) was used to run quantitative real-time PCR (qRT-PCR) with LightCycler^®^ Fast Start DNA Master PLUS SYBRGreen I (Roche, cat. #03515869001) and specific intron-spanning primers (Table 2 below).

Analysis was performed using the relative quantification (RQ) method and normalization versus housekeeping genes, with *18S rRNA* and *GAPDH* serving as endogenous controls.

## Figures and Tables

**Figure 1 ijms-25-08980-f001:**
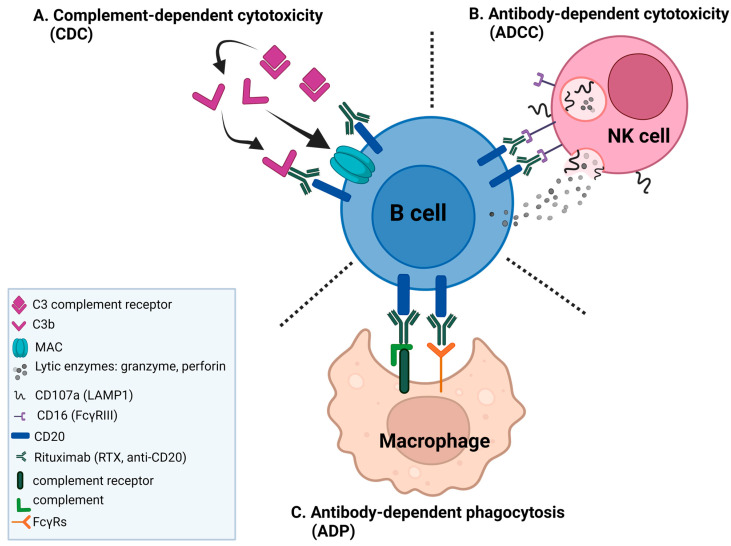
Mechanisms of rituximab-mediated cell death. RTX-coated malignant B cells are eliminated by three major mechanisms. (**A**) binding of RTX to CD20 on the B cell surface causes the activation of the complement cascade, which generates the membrane attack complex (MAC), directly inducing target cell lysis by complement-dependent cytotoxicity (CDC). (**B**) The binding of RTX to the target cell allows interaction with natural killer (NK) cells via the Fcγ receptor (FcγR) IIIa (CD16), which leads to antibody-dependent cellular cytotoxicity (ADCC). (**C**) The Fc portion of RTX and the deposited complement fragments allow recognition by both FcγRs and complement receptors on macrophages, which leads to antibody-dependent phagocytosis (ADP). Generated with BioRender.com, accessed on 9 July 2024.

**Figure 2 ijms-25-08980-f002:**
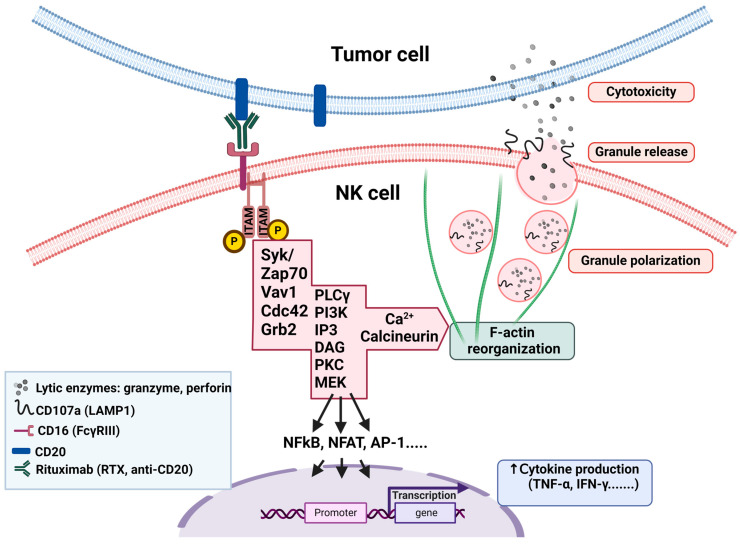
Scheme illustrating the CD16 downstream signaling cascades in NK cells, which lead to cytotoxicity against RTX-coated target cells and cytokine production. The binding of CD16 to the Fc of RTX on CD20-positive target cells initiates a complex sequence of events through ITAM phosphorylation, which induces an activation cascade of Syk family kinases, PLCγ, and subsequently increases the intracellular Ca^2+^ levels, and calcineurin activation. This signaling cascade results in the rapid mobilization of granules to the immunologic synapse of NK with the target cell and merges with the plasma membrane. In the course of this process, the movement of granules containing granzymes and perforin is facilitated by CD107a/LAMP1, which appears on the cell surface. At this stage, degranulating NK cells, considered as fully activated NK cells, can be identified using flow cytometry analysis and CD107a-specific antibodies. The ITAM/SyK/PLCγ/Ca^2+^/calcineurin signaling axis also promotes the activation of transcription factors NFkB, NFAT, and AP-1 for enhancing the production of proinflammatory cytokines (such as TNFα and IFNγ), which play a role in engaging other immune cells for the anti-tumor immune response. Generated with BioRender.com, accessed on 9 July 2024.

**Figure 3 ijms-25-08980-f003:**
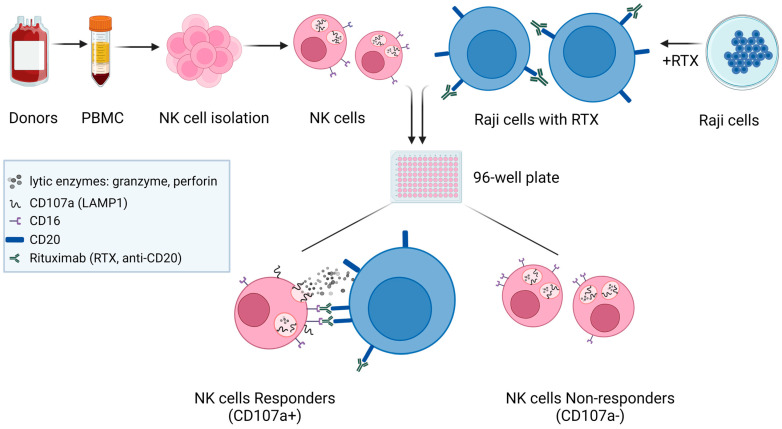
Experimental design of CD16-mediated NK cell degranulation. Scheme explaining the design of experiments leading to the degranulation of NK cells. The buffy coats from human donors were used to isolate PBMCs and NK cells sequentially. The Burkitt lymphoma cell line, Raji, was preincubated with the therapeutic anti-CD20 antibody, rituximab (RTX), followed by co-incubation with isolated primary NK cells. After co-incubation, two populations of NK cells can be identified using flow cytometry—responders (cells positive for degranulation marker, CD107a/LAMP1) and non-responders (NK cells CD107a-negative). Generated with BioRender.com, accessed on 9 July 2024.

**Figure 4 ijms-25-08980-f004:**
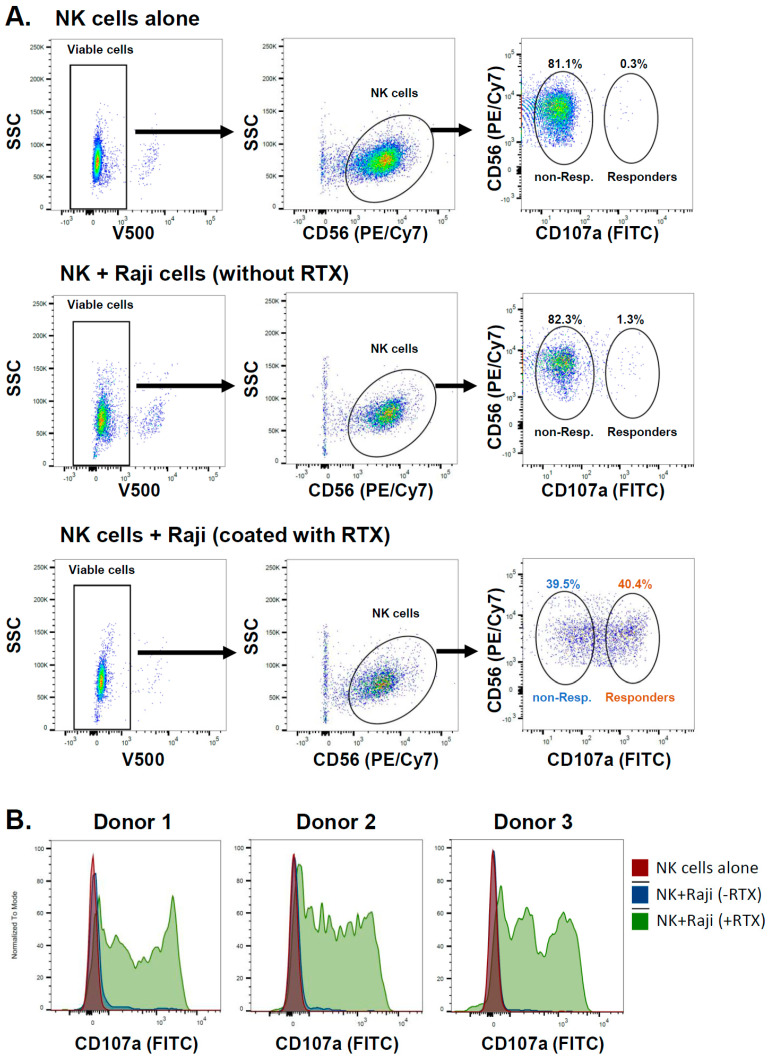
Primary NK cells exposed to Rituximab-coated Raji cells degranulate non-uniformly. (**A**) Examples of graphs generated using the FlowJo software (v10.7.1) depicting the flow cytometry analysis of primary NK cell degranulation. Dot plots with SSC and V500 channels (left panels) allowed the discrimination of viable cells (FVS510-negative events). Dot plots with SSC and PE/Cy7 channels (middle panels) enabled the detection of the NK cell population (CD56-PE/Vio770-positive events), followed by an analysis of NK cells for the presence of CD107a-FITC signal (right panels). The samples of NK cells alone (top row) exhibited the CD107a-negative events (non-responders) exclusively. The samples of NK cells co-incubated with Raji cells, without RTX (middle row), also showed the presence of mainly CD107a-negative events (non-responders). However, the samples of NK cells co-incubated with RTX-coated Raji cells (bottom row) exhibited both the CD107a-negative (non-responders) and the CD107a-positive (responders) populations. (**B**) Examples of graphs generated using the FlowJo software depicting the histogram plots of CD107a analysis in NK cells isolated from three healthy donors. Only the co-incubation with RTX-coated Raji cells induced the appearance of CD107a-positive events in NK cell populations.

**Figure 5 ijms-25-08980-f005:**
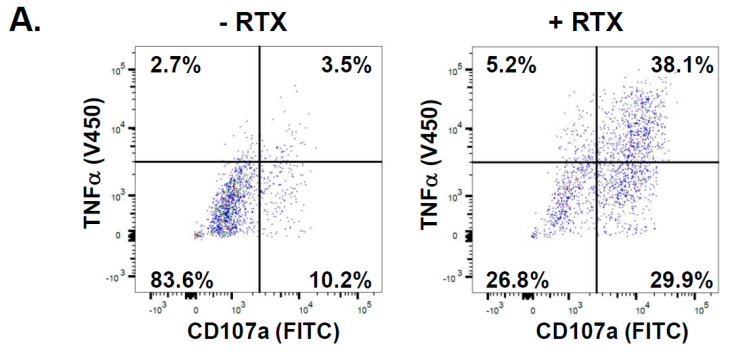
The degranulating population of NK cells produces increased levels of TNFα. (**A**) Examples of graphs generated using the FlowJo software depicting the flow cytometry analysis and percentage of NK cells exhibiting low and high intracellular TNFα staining in the non-responder (CD107a-negative events) and responder populations (CD107a-positive events) of NK cells co-incubated with Raji in the absence of RTX (**left** panel) and in the presence of RTX (**right** panel). (**B**) The percentage of cells with low and high TNFα staining in NK cells deriving from three donors was summarized in the tables. (**C**) The graph presents the mean fluorescence intensity (MFI) of TNFα staining in the non-responders and the responders populations of NK cells co-incubated with Raji without RTX (−RTX) or with RTX-coated Raji (+RTX) ** <0.005; **** < 0.0001.

**Figure 6 ijms-25-08980-f006:**
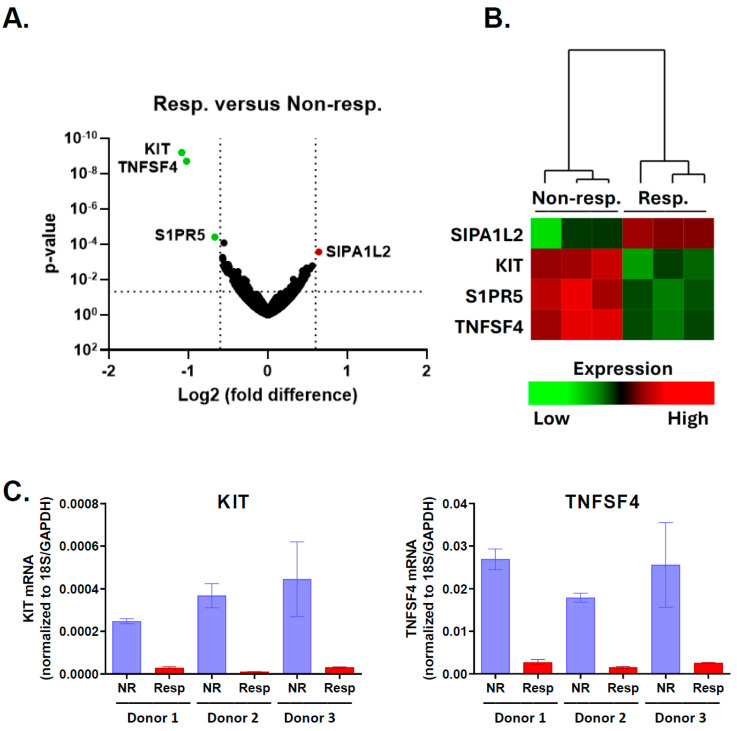
The expression of a small number of genes is significantly changed in response to CD16 stimulation. (**A**) The volcano plot compares gene expression in the responder versus the non-responder groups of NK cells. Green dots represent genes with significant down-regulation (log2 fold change < −0.6), while the red dot represents a significantly up-regulated gene in responder NK cells. (**B**) The heatmap presents the hierarchical clustering of mRNAs exhibiting significant (*p*-value < 0.05) changes between the non-responder (“Non-resp.”) and responder (“Resp.”) populations. (**C**) The qRT-PCR analysis of *KIT* mRNA levels (left panel) and *TNFSF4* mRNA levels (right panel) in sorted non-responder (NR) and responder (Resp) populations of primary NK cells deriving from healthy donors (Donor 1, Donor 2, and Donor 3). The Y axis presents the “Target per Reference” expression values, therefore providing normalization to reference genes, namely 18S rRNA and GAPDH.

**Table 1 ijms-25-08980-t001:** The percentage of non-responder (non-degranulating) and responder (degranulating) NK cells identified during flow cytometry analysis based on CD107a-positive events among CD56-positive events (NK cells).

	Non-Responders (% Parent)	Responders (% Parent)
Donor 1	39.5	40.4
Donor 2	38.2	45.1
Donor 3	38.3	41.2
Donor 4	50.5	24.7
Donor 5	48.3	34.4
Donor 6	23.7	49.2

**Table 2 ijms-25-08980-t002:** Primers used in qRT-PCR.

Target	Forward Primer	Reverse Primer
*KIT*	ACCAACACCGGCAAATACA	AAGCTTGGCAGGATCTCTAAC
*TNFSF4*	TTCCAACTGAAGAAGGTCAG	GAAGTCATCCAGGGAGGTAT
*GAPDH*	AGCCACATCGCTCAGACAC	GCCCAATACGACCAAATCC
*18S rRNA*	GCAATTATTCCCCATGAACG	GGGACTTAATCAACGCAAGC

## Data Availability

Data is contained within the article or Appendix A.

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
