# Peer review of "NK Cell Degranulation Triggered by Rituximab Identifies Potential Markers of Subpopulations with Enhanced Cytotoxicity toward Malignant B Cells"

_ijms, 2024, doi:10.3390/ijms25168980_

Round 1

Reviewer 1 Report

Comments and Suggestions for Authors

Monoclonal antibodies like rituximab, which targets CD20, induce NK cell-mediated ADCC and are used to treat B-cell malignancies such as non-Hodgkin lymphoma, albeit with varying success. This study aimed to analyze the gene expression of NK cells during the cytolytic response triggered by rituximab. NK cells were co-cultured with rituximab-coated Raji cells and sorted based on CD107a, a degranulation marker. RNA-seq revealed that KIT and TNFSF4 genes were significantly down-regulated in degranulating NK cells. Based on these data, the authors suggest that targeting c-KIT and OX40L through pharmacological inhibition or gene ablation could enhance NK cell anti-tumor activity in response to rituximab.

Issues to address:

-Figure 2.  Provide a histogram plot summarizing all the data and with appropriate statistics. Also, provide a label to each panel so that the reader can understand the figure without the need of the legend.

-Figure3. The histogram plot should include the data relative to the four different populations obtained with the gating strategy.

-Figure 4. Panel A could be provided only as supplementary file.

-The RNA-seq analysis is the most problematic point in this paper. A lot of information is missing. How many replicates have been performed? Was the entire transcriptome analyzed? Which sequencer was used for the analysis? How were the data preprocessed and normalized? Why the author did not use FDR for the statistical analysis? How was the differential expression analysis performed? Have the author uploaded the RNA-Seq data to publicly-available repositories, such as GEO?

Reviewer 2 Report

Comments and Suggestions for Authors

This study analyzed the gene expression of NK cells after rituximab Fc engagement on CD16. Based on CD107a expression the cells were classified into responders and non-responders. Among all the transcriptome, 4 genes were identified, and 2 of them (KIT and TNFSF4) confirmed using RTPCR. The manuscript is well written, it is easy to read and to understand. Of note, in the context of B lymphocytes and rituximab, other mechanims of action may be involved (not only NK cells). The results are worth publishing.

 Additional comments:

 (1) Line 40. Regarding Rituximab. Could you please explain the different mechanism of action on the B lymphocytes?

 (2) Line 50. The interaction of rituximab antibody and CD16 on NK cells. Could you please show a figure with this and the other mechanisms?

 (3) Line 51. M1 macrophages can also express CD16. Is this correct? In that case, Th1, M1, NK, and Tc response is one of the mechanisms of RTX-based therapy? However, in the tissue of DLBCL, NK cells are very few (?).

 (4) Lines 54-68. Please show the intracellular pathway using a figure.

 (5) Line 71. Please describe in more detail the mutations of CD20 present in DLBCL and their gain/loss of function.

 (6) Line 83. Regarding reference 18. Could you please provide more details? What immune-checkpoints and what mechanism?

 (7) Line 86. Regarding “Understanding the mechanisms behind NK cell response to RTX to overcome resistance is crucial for maximizing mAbs therapeutic benefits”. Do you refer only of resistance of RTX mechanism on the NK cells, or resistance in general of DLBCL?

 (8) In Figure 1. CD16 is interacting with Fc of Rituximab? How CD16 “interacts”/”pathways” to CD107a?

 (9) The official gene name of CD107a is LAMP1 (Lysosomal Associated Membrane Protein 1). Please also add it in legend of Figure 1. You may also add “Mechanistically, participates in cytotoxic granule movement to the cell surface and perforin trafficking to the lytic granule (PubMed:23632890).”

 (10) Is CD107a detected by flow cytometry only on the surface of the NK cells?

 (11) Line 145. Please explain briefly how the non-responder and responder cells were separated.

 (12) Line 152. Please add the RNA-sequencing technique. Is it all transcriptome (it may be explained at the end of the manuscript, but better to explain to reader in this section).

 (13) In the volcano plot, what is the criteria for the cutoff to identify the relevant genes?

 (14) Regarding the volcano plot, are 3 cases enough for the statistical analysis?

 (15) Line 251. What is the function of KIT and OX40L (TNFSF4) in the degranulation of NK cells? Should responder NK be the activated/functiontal cells?

 (16) Does the inhibition of KIT and TNFSF4 leads to a change of phenotype from non-responder to responder?

 (17) Are KIT and TNFSF4 immuno-oncology markers?

 (18) Any further information regarding SIPA1L2?

 (19) Are responders NK cells the ones that have anti-tumor activity? Could it be that the non-responders are also relevant for the anti-cancer response but need a different ligand/stimulus?

Round 2

Reviewer 1 Report

Comments and Suggestions for Authors

Issues addressed